# Weight Gain During Pregnancy Following Bariatric Surgery: Exploring the Influence of Weight Stability and Surgery-to-Conception Interval

**DOI:** 10.3390/jcm14134666

**Published:** 2025-07-01

**Authors:** Taylor M. Guthrie, Sandra Lee, Alka Kothari, William Pinzon Perez, Sailesh Kumar, Helen Truby, Susan de Jersey

**Affiliations:** 1Faculty of Health, Medicine & Behavioural Sciences, University of Queensland, St Lucia, QLD 4072, Australia; alka.kothari@health.qld.gov.au (A.K.); sailesh.kumar@mater.uq.edu.au (S.K.); h.truby@uq.edu.au (H.T.); susan.dejersey@health.qld.gov.au (S.d.J.); 2Dietetics and Foodservices, Royal Brisbane Women’s Hospital, Herston, QLD 4029, Australia; 3Maternity Services, Caboolture Hospital, Caboolture, QLD 4510, Australia; sandra.lee@health.qld.gov.au; 4Obstetrics and Gynecology, Redcliffe Hospital, Redcliffe, QLD 4020, Australia; 5Research and Statistical Support Service, Faculty of Medicine, Provided by QCIF Facility for Advanced Bioinformatics (QFAB), University of Queensland, Brisbane, QLD 4072, Australia; w.pinzonperez@qcif.edu.au; 6Mater Research Institute, University of Queensland, South Brisbane, QLD 4101, Australia

**Keywords:** obesity, bariatric surgery, preconception, gestational weight gain

## Abstract

**Background/Objectives**: Weight loss following bariatric surgery can improve fertility. Current guidelines recommend delaying pregnancy for at least 12 months post-surgery for weight stabilization and to support healthy gestational weight gain (GWG). However, evidence supporting this recommendation is limited. This study investigated the impact of preconception weight stability and the surgery-to-conception interval on GWG and examined risk factors for GWG above or below recommendations. **Methods**: Women aged 18–45 years with singleton pregnancies post-bariatric surgery were recruited before 23 weeks’ gestation and followed until delivery. Participants self-reported their weight for the 6 months preceding conception and again at 36 weeks’ gestation via an online survey. Weight change (as a percentage of pre-pregnancy weight) was analyzed using stepwise linear and multivariate logistic regression. **Results**: Sixty-nine participants completed the study. The percentage of body weight change in the 6 months before conception ranged from −34% to +21%, with significantly greater preconception weight loss observed in those who conceived within 12 months of surgery (*p* < 0.001). The pre-pregnancy BMI and preconception weight change together explained 24% of the variation in GWG (*p* < 0.001), while the surgery-to-conception interval was not a significant predictor (*p* = 0.502). While 70% (34/49) of participants experienced weight gain outside of recommendations, no significant risk factors could be identified. **Conclusions**: Weight trajectory prior to conception is a key factor to predict GWG rather than the surgery–conception interval. These findings have important implications for family planning and clinical guidance following bariatric surgery.

## 1. Introduction

Bariatric surgery has become increasingly prevalent since the early 2000s and is now established as a highly effective long-term treatment for obesity [1,2]. In many developed countries, including Australia and the United Kingdom, women of childbearing age constitute the largest demographic undergoing bariatric surgery [3,4]. Despite recent advances in pharmacotherapy for the treatment of obesity, bariatric surgery remains a highly effective fertility enhancing option [5,6]. Benefits of bariatric surgery include reduced incidences of gestational diabetes mellitus, pregnancy-induced hypertension, pre-eclampsia, and large-for-gestational-age or macrosomic infants [7,8]. However, it is also associated with maternal micronutrient deficiencies, small-for-gestational-age or growth-restricted infants, and an increased likelihood of preterm birth [7,8].

International guidelines recommend that conception be delayed until after the period of rapid and acute weight loss, usually 12 to 24 months post-bariatric surgery [9,10]. This interval allows maximal weight loss to be achieved [10] and the stabilization of preconception weight [11,12]. However, evidence is limited regarding whether delaying pregnancy results in weight stability or improved outcomes. Post-operatively, many women remain above a healthy body mass index (BMI) [13,14,15] and 20 to 40% regain weight after 12 to 18 months [16,17]. Therefore, delaying pregnancy may lead to women commencing pregnancy at a higher BMI, potentially increasing their risk of pregnancy complications [18]. Conversely, as weight reduction is most rapid in the first year following bariatric surgery, there may be a risk of inadequate gestational weight gain (GWG) in women who fall pregnant within 12 [19] or 18 months [20] of surgery.

Informed by this background, the aim of this study was to explore the impact of weight change 6 months prior to conception and the surgery-to-conception interval on GWG in women with a history of bariatric surgery. A secondary aim was to identify risk factors for GWG above or below recommendations. This analysis is nested within a research project examining maternal dietary intake and micronutrient status throughout pregnancy after bariatric surgery, and the participants are part of a prospective cohort study examining maternal nutrition during pregnancy after bariatric surgery [21].

## 2. Materials and Methods

This prospective observational study was conducted across four hospital sites in Queensland, Australia. Ethical approval was provided by the Metro North Hospital and Health Services Human Research and Ethics Committee (EC00168) and was registered with the Australia and New Zealand Clinical Trials Registry [registration no. ACTRN1263000495628]. The study was conducted in accordance with the declaration of Helsinki. Recruiting hospitals were all publicly funded, with three secondary referral hospitals (each with 1500–2000 births annually) and one quaternary referral hospital (5000 births annually). Recruitment commenced in June 2022 and concluded in February 2024.

Pregnant women aged 18–45 and at <23 weeks gestation with a history of bariatric surgery (gastric band, sleeve, bypass, or revisional procedure) were eligible. Exclusion criteria included multiple pregnancies, conditions affecting micronutrient metabolism (renal failure, inflammatory bowel disease), Type 1 Diabetes Mellitus, or inability to provide consent.

Initial study information was provided to women by their midwife, dietitian, obstetrician, or obstetric physician attending any of the four public antenatal clinics. Those who expressed interest in participating were contacted individually by the lead researcher to confirm eligibility and enrolled after providing full informed written consent. Participating women had access to standard antenatal and dietetic care in accordance with local practice guidelines [22]. This included modified screening for gestational diabetes mellitus, regular checks for maternal micronutrient status, and the monitoring of fetal growth via ultrasound.

Participants completed an online questionnaire at enrolment (<23 weeks), which included demographic data, height, self-reported weight (pre-operatively, 3 and 6 months before conception, as well as immediately preconception), date, and type of bariatric surgery procedure. GWG was self-reported at 36 weeks’ gestation. The interval between surgery and conception was categorized as either <12 months or ≥12 months. Participants were categorized by the type of bariatric surgery: restrictive (gastric band, sleeve gastrectomy) or malabsorptive (gastric bypass). Maternal BMI was calculated using weight before bariatric surgery and weight immediately before pregnancy. Weight change 3 and 6 months preconception was examined as a percentage of weight immediately prior to conception (referred to as ‘pre-pregnancy weight’) to account for a different baseline BMI among participants. This was determined using the following equation:$$\left(\frac{\mathrm{Pre-pregnancy}\;weight-weight\;3\;or\;6\;months\;preconception}{\mathrm{Pre-pregnancy}\;weight\;}\right)\times 100$$

Weight stability was defined as within ±5% of pre-pregnancy weight. Participants were defined as gaining weight if their weight change exceeded 5% and losing weight if their weight change was greater than −5% compared to their pre-pregnancy weight. Participants’ GWG was assessed according to the Institute of Medicine [11] guidelines. As pre-pregnancy BMI influences healthy pregnancy weight gain, GWG was analyzed as % pre-pregnancy weight rather than an absolute value. This was calculated using the following equation:$$\left(\frac{Weight\;at\;36\;weeks-\;\mathrm{Pre-pregnancy}\;weight\;}{\mathrm{Pre-pregnancy}\;weight\;}\right)\times 100$$

Data analysis was performed using the Statistical Package for the Social Sciences (SPSS version 30, 2024, IBM Corp. Armonk, NY, USA). The distribution of continuous variables was assessed using Shapiro–Wilk test. Normally distributed data were summarized as a mean ± standard deviation (SD), while non-normally distributed data were presented as a median with interquartile range (IQR). Comparisons between continuous and categorical variables were performed using independent samples t-tests and ANOVAs (normally distributed variables), and Mann–Whitney U and Kruskal–Wallis H tests (non-normally distributed variables). Spearman’s correlation was conducted to compare relationships between continuous variables: preconception weight change, the surgery-to-conception interval, and GWG. A stepwise linear regression explored whether weight change 6 months preconception and the surgery-to-conception interval were independent predictors of weight change during pregnancy (as % pre-pregnancy weight), when maternal age, parity, pre-pregnancy BMI, and bariatric surgery type were included in the model. The criteria for variable entry and removal were set at a probability-of-F-to-enter ≤0.05, probability-of-F-to-remove ≥0.10. The assumptions required for linear regression—linearity, independence of errors, homoscedasticity, normality of residuals, and absence of multicollinearity—were examined using residual plots, tolerance, and variance inflation factors, respectively. To prevent overfitting, the adjusted R^2^ value is reported. A multivariate binary logistic regression analysis examined whether weight change 6 months preconception and the surgery-to-conception interval impacted the odds ratio (OR) of GWG outside of recommendations (gaining more or less weight than recommended based on pre-pregnancy BMI) when controlling for confounders (maternal age, parity, pre-pregnancy BMI, and bariatric surgery type). Multicollinearity was checked using the variance inflation factor, outliers were identified using the Mahalanobis distance, and linearity was examined using log transformation in logistic regression. A *p*-value of <0.05 was considered statistically significant.

## 3. Results

Sixty-nine women participated in the study (see Figure 1). Participant characteristics are reported in Table 1. 

### 3.1. Preconception Weight

Weight change prior to conception varied between participants (see Table 2 for anthropometric longitudinal measurements). Three months prior to conception, 80% (47/59) of women were defined as weight stable, 15% (10/59) were losing weight, and 3% (n = 2/59) were gaining weight. Six months before conception, 47% (30/64) were weight stable, 33% (23/64) were losing weight, and 16% (11/64) were gaining weight. Women with a surgery-to-conception interval of <12 months had a negative median weight change at 3 months (−10.1% (8) vs. 0.0% (4), *p* < 0.001) and 6 months prior to conceiving (−16.9% (15) vs. 0.0% (4), *p* < 0.001) indicating weight loss leading up to conception.

### 3.2. The Impact of Weight Change Preconception and Gestational Weight Gain

Women gained on average 7.3 ± 7.3 kg during pregnancy, comprising 9.6 ± 9.4% of their pre-pregnancy body weight. GWG was similar between women who had undergone restrictive and malabsorptive procedures (*p* = 0.831). Thirty-one percent (15/49) of participants achieved recommended GWG during pregnancy, 33% (16/49) gained less weight than recommended, and 37% (18/49) gained more than recommended. Women who exceeded GWG recommendations were more likely to be gaining weight 3 and 6 months before conception than those who met GWG recommendations or those with inadequate GWG (*p* = 0.029, *p* = 0.025, respectively, see Figure 2).

Women with a surgery-to-conception interval of <12 months had a lower mean GWG during pregnancy (2.4 ± 5.7%) compared to those who conceived after 12 postoperative months (11.0 ± 9.4%, *p* = 0.016). However, there were no significant differences between the surgery-to-conception interval and GWG categories (*p* = 0.307). The surgery-to-conception interval was modestly and positively correlated with weight change (R = 0.393, *p* = 0.002) 3 and 6 months preconception (R = 0.487, *p* < 0.001) but not GWG (R = 0.139, *p* = 0.331).

Stepwise linear regression identified significant predictors of GWG. In the first step, pre-pregnancy BMI was statistically significant in the model *F* (1, 47) = 10.20, *p* = 0.003. In the second step, weight change 6 months preconception was also included in the model *F* (2, 46) = 8.70, *p* < 0.001. The final model, including pre-pregnancy BMI and weight change 6 months preconception, accounted for a moderate proportion of the variation in GWG (R^2^ = 0.274, adjusted R^2^ = 0.243). Increasing pre-pregnancy BMI (*t =* −3.56, *p* < 0.001, *β* = −0.574) was associated with a reduction in GWG, whereas increasing weight gain 6 months preconception (*t* = 2.469, *p* = 0.017, *β =* 0.297) was associated with increased GWG. Maternal age at conception (*t =* 0.009, *p* = 0.993), parity (*t =* −1.382, *p* = 0.174), and the surgery-to-conception interval (*t =* 0.685, *p* = 0.497) were excluded from the model.

To further explore the impact of weight change preconception, univariate analysis demonstrated that weight change 6 months preconception did not impact the odds of GWG which was outside recommended parameters (X^2^ = 0.000, *p* = 0.994, OR 1.00, 95% CI 0.94–1.07). This finding remained consistent after adjusting the model to account for pre-pregnancy BMI and surgery type (adjusted OR 0.980, 95% CI 0.91–1.05, *p* = 0.577). Parity (*p* = 0.824), maternal age (*p* = 0.408), and the surgery-to-conception interval (*p* = 0.800) did not significantly affect the odds of GWG outside recommendations during univariate analysis and were therefore excluded from multivariate logistic regression.

## 4. Discussion

This study compared the impact of the surgery-to-conception interval and preconception weight change on GWG in women post-bariatric surgery. It revealed that the proportion of women who were weight stable 6 months and 3 months preconception were ~50% and ~80%, respectively. Delaying pregnancy by at least 12 months was found to reduce weight change preconception, but it did not influence GWG. Only preconception weight change and BMI were identified as independent predictors of GWG. Importantly, ~70% of participants had a GWG outside recommended parameters (33% below recommendations and 37% above recommendations). No significant predictors of this outcome could be identified. This highlights the complexity of GWG post-bariatric surgery, suggesting that while preconception weight change may play a role, other factors, currently unknown, are likely to impact a woman’s ability to achieve their recommended GWG.

Previous studies [15,23,24] suggest that conceiving within 12 months of bariatric surgery increases the risk of inadequate GWG, while delaying pregnancy by 12 [15] or 18 [25] months may increase excess GWG. However, these studies overlook the potential role of preconception weight change. Weight loss following bariatric surgery varies widely, some individuals regain weight within 6 months and others continue to lose weight 3 years after surgery [26]. In this study, the surgery-to-conception interval affected preconception weight change, consistent with prior findings [27], but multivariate analysis showed preconception weight change had a stronger influence on GWG. Optimizing GWG is associated with reduced rates of GDM, hypertensive disorders, caesarian birth, abnormal neonatal growth, and preterm birth [28], as well as improved long-term metabolic health for the mother and child [29]. While limited, research suggests pregnancy may not affect long-term post-surgical weight loss [30,31], though excess GWG and postpartum weight retention are more common in those who delay pregnancy following bariatric surgery [31]. Additionally, qualitative studies highlight the emotional burden of weight changes during pregnancy post-bariatric surgery and a need for better support [32]. Monitoring preconception weight trends may help identify risks for suboptimal GWG and guide individualized care.

Our finding that women who were losing weight preconception were predisposed to insufficient GWG has implications for clinical practice. Inadequate GWG may reflect insufficient energy intake, more pronounced nutrient malabsorption, or bariatric surgery-related symptoms [33,34]. Furthermore, women who are stable or gaining weight at conception appear more likely to experience GWG above recommendations. Our results suggest that these groups of women may have different needs when it comes to nutrition-related care. Existing guidelines recommend individualized nutrition therapy for all women post-bariatric surgery ideally prior to and throughout pregnancy [35]. However, limited evidence is available to guide specific nutrition therapy interventions.

Outside of bariatric surgery, dietary interventions embedded within a behaviour change framework that address energy and macronutrient intake and/or improve dietary patterns are effective and reduce overall GWG [36] and weight retention 6 months postpartum [37]. Conversely, interventions for women at risk of insufficient GWG that focus on adequate energy and micronutrient intake, as well as address barriers to intake (such as nausea and vomiting), can improve outcomes [33]. However, such interventions are yet to be tested in a population post-bariatric surgery and existing qualitative data suggests that dietary interventions seldom meet pregnant women’s needs post-bariatric surgery [32]. Preconception weight change and BMI was only able to predict 24% of the variation in GWG in this study. This suggests that a large proportion of GWG is influenced by other factors. Evidence suggests that weight change during pregnancy is regulated by a complex interplay of modifiable (dietary intake, physical activity) and non-modifiable factors (genetics, ethnicity, socioeconomic status, and mental health) [36]. These factors require further examination alongside preconception weight status to develop a more comprehensive understanding of predictors of GWG post-bariatric surgery and inform person-centred interventions.

The strengths of this study include its prospective nature which limits the impact of recall bias on data such as pre-pregnancy weight and GWG. However, it is important to acknowledge that weight stigma and recall bias may influence the accuracy of self-reported weight in populations with obesity [38]. There was insufficient evidence available to inform a power calculation, therefore our study may be underpowered to detect changes in adherence to GWG recommendations. Additionally, combining women with GWG below and above recommendations may introduce bias because of their very different metabolic profiles. This may hide important results, such as risk factors for excess GWG, that do not present in women with inadequate GWG. However, this approach was necessary due to the relatively small sample size and the variation in weight profiles. The sample size also prohibited the inclusion of other potential confounders influencing GWG such as dietary behaviour, socioeconomic status, and physical activity. This presents another limitation, though some of this data has been published previously [21]. Collectively these factors may explain the adjusted R^2^ of 0.243, which suggests that a large proportion of GWG is not explained by preconception weight change and BMI. As this study did not assess non-linear relationships or interaction effects, it may have overlooked other non-linear factors that contribute to GWG. This study was conducted primarily on gastric sleeve recipients, who have been underrepresented in previous work of this nature [15,25]. However, this may limit generalizability to gastric band, bypass, and other bariatric surgery recipients seeking preconception care.

## 5. Conclusions

This analysis highlights the importance of preconception weight trajectory, rather than the surgery-to-conception interval, as influential in GWG following bariatric surgery. It identifies that a significant proportion of women experience both insufficient and excessive GWG, which may have short- and long-term consequences for both the mother and child. If these findings are replicated in larger, more diverse populations, this may warrant a shift from the general recommendation of a time-delineated delay in conception to a more tailored, individualized approach, providing preconception guidance that reflects each woman’s unique weight loss journey.

## Figures and Tables

**Figure 1 jcm-14-04666-f001:**
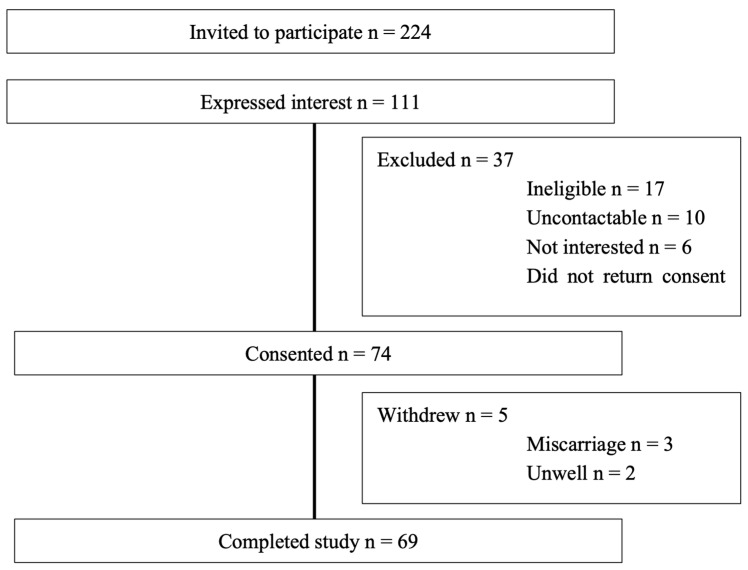
Participant Flow.

**Figure 2 jcm-14-04666-f002:**
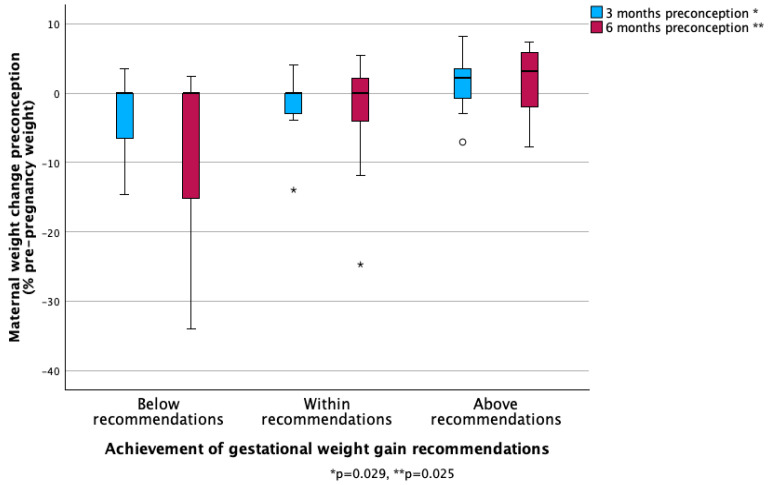
Preconception weight change against Institute of Medicine’s gestational weight gain recommendations.

**Table 1 jcm-14-04666-t001:** Characteristics of study participants.

	Median (IQR)	Range
Maternal age at conception (years)	31.2 ± 4.8 *	22–42
Parity	1 (2)	0–4
Ethnicity, % (n=)		
Caucasian	78 (53/68)	
Aboriginal and/or Torres Strait Islander	10 (7/68)	
Pacific Islander	3 (2/68)	
Other	9 (6/68)	
Bariatric surgery-to-conception interval (months)	30 (51)	0–186
<12 months, % (n=)	19 (13/68)	
Bariatric Surgery type, %, (n=)		
Gastric band	1 (1/69)	
Gastric sleeve	68 (47/69)	
Gastric bypass	30 (21/69)	
Revisional surgery % (n=)	7 (5/69)	
Education attainment, % (n=)		
Postgraduate or undergraduate degree	24 (16/68)	
Trade, technical certificate, or diploma	40 (27/68)	
Completed schooling	28 (19/68)	
Did not complete schooling	9 (6/68)	

* Mean ± SD.

**Table 2 jcm-14-04666-t002:** Anthropometric measures prior to and throughout pregnancy.

	Median (IQR)	Range
Pre-bariatric surgery BMI (kg/m^2^)	45.1 (13.0)	32.7–73.1
Weight change 3 months preconception (% pre-pregnancy weight) ^I^	0 (6)	−15–8
Gaining weight (2/59)	6.8 ^II^	
Losing weight (10/59)	−10.3 (6)	
Weight change 6 months preconception (% pre-pregnancy weight) ^III^	0 (11)	−34–21
Losing weight (23/64)	−10.0 (12)	
Gaining weight (11/64)	7.0 (2)	
Pre-pregnancy BMI (kg/m^2^)	30.4 (7.3)	18.4–50.4
Pre-pregnancy BMI Category, % (n=)		
BMI < 18.5	2 (1/66)	
BMI 18.5–24.9	13 (9/66)	
BMI 25–29.9	25 (18/66)	
BMI > 30	57 (38/66)	
Gestational weight gain, (kg) ^IV^	7.3 ± 7.3 *	−12–22
% pre-pregnancy weight	9.6 ± 9.4 *	−10–37
Gestational weight gain alignment with recommendations, % (n=) ^IV^		
Below recommendations	33 (16/49)	
Within recommendations	31 (15/49)	
Above recommendations	37 (18/49)	

* Mean ± SD. ^I^ Data not available for 10/69. ^II^ Insufficient data for IQR. ^III^ Data not available for 4/69. ^IV^ 20/69 unsure or gave birth before responding.

## Data Availability

The data presented in this study are available by request only from the corresponding author due to privacy and ethical reasons.

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
