# Peer review of "Weight Gain During Pregnancy Following Bariatric Surgery: Exploring the Influence of Weight Stability and Surgery-to-Conception Interval"

_jcm, 2025, doi:10.3390/jcm14134666_

Round 1

Reviewer 1 Report

Comments and Suggestions for Authors

Congratulations on this topic seeking for a better, more tailored recommendation pregnancy after bariatric surgery.

The main subject of the research is to define whether established time lapse after bariatric surgery or weight stability is more important regarding gestational weight gain and complications arising from it.

Several reports and studies are referring to this topic, but only a few have been designed to debate a recommendation on the best timing for pregnancy after bariatric surgery. The research challenges the actual recommendation on timing after bariatric procedures to a personalized indication depending on many factors.  It is expressed that many other variables are not taken into account in their research; however, this is properly addressed in the paper.   Methodology, references, and conclusions seem right and according to data shown in the manuscript.

Author Response

Please see attached response to reviewer feedback. 

Reviewer 2 Report

Comments and Suggestions for Authors
  1. This study did not explain how the sample size was calculated. Please add a power calculation or explain that the study may be underpowered and could have missed small or medium effects. If you cannot fix this now, add it as a limitation in the end of the discussion.
  2. The weight data was self-reported and not confirmed with medical records. This can be wrong because of memory problems or body image concerns. Please say clearly in your methods and discussion that this is a measurement limitation.
  3. The final regression model only explained about 24% of the difference in weight gain. That means other important things are missing in the model. Please explain this in the discussion and mention that other factors like diet, activity or mental health may be important but were not measured.
  4. You used one model for women who gained too much or too little weight. But these are different problems and may have different causes. Please say in the discussion that combining them may hide important results, and this is a limitation.
  5. Most women had gastric sleeve surgery, but the study did not look closely at differences between surgery types. Please add a comment in the discussion that this limits the ability to make suggestions for other surgery types like bypass.
  6. You did not ask about mental health, eating behaviors or emotional stress. These things can affect weight gain, especially in women after bariatric surgery. Please write in the discussion that this was not included and should be studied in the future.
  7. You did not include data on the women’s diets, exercise, or social and financial situation. These things can affect both weight and pregnancy. Please say that this is a limitation, even if some of this was studied before in another paper.
  8. The study conclusion says doctors should change their advice to patients. But the results are early and should be tested more. Please change the conclusion to be more careful. You can say the findings are interesting and should be explored in larger studies, but do not suggest practice change yet.
  9. The adjusted R² of 0.243 in your regression indicates most variance in GWG is unexplained. Why did you not test for interaction terms or non-linear effects? You should discuss this statistical modeling limitation.

Author Response

Please see attached our response to reviewer feedback. 

Round 2

Reviewer 2 Report

Comments and Suggestions for Authors

All the review points I asked were answered well and added as study limitations. Now the paper has a more honest and careful scientific structure.